# β₃-Adrenoceptor redistribution impairs NO/cGMP/PDE2 signalling in failing cardiomyocytes

Sophie Schobesberger[1,2†], Peter T Wright[1†], Claire Poulet[1], Jose L Sanchez Alonso Mardones[1], Catherine Mansfield[1], Andreas Friebe[3], Sian E Harding[1], Jean-Luc Balligand[4], Viacheslav O Nikolaev[2‡*], Julia Gorelik[1‡*]

[1]Myocardial Function, National Heart and Lung Institute, Imperial College London, ICTEM, Hammersmith Hospital, London, United Kingdom; [2]Institute of Experimental Cardiovascular Research, University Medical Center Hamburg-Eppendorf, German Center for Cardiovascular Research (DZHK) partner site Hamburg/Kiel/Lübeck, Hamburg, Germany; [3]Physiologisches Institut, University of Würzburg, Würzburg, Germany; [4]Pole of Pharmacology and Therapeutics (FATH), Institut de Recherche Expérimentale et Clinique (IREC), Université Catholique de Louvain (UCLouvain), Brussels, Belgium

*For correspondence:
v.nikolaev@uke.de (VON);
j.gorelik@imperial.ac.uk (JG)

†These authors contributed equally to this work
‡These authors also contributed equally to this work

Competing interests: The authors declare that no competing interests exist.

**Abstract** Cardiomyocyte β₃-adrenoceptors (β₃-ARs) coupled to soluble guanylyl cyclase (sGC)-dependent production of the second messenger 3',5'-cyclic guanosine monophosphate (cGMP) have been shown to protect from heart failure. However, the exact localization of these receptors to fine membrane structures and subcellular compartmentation of β₃-AR/cGMP signals underpinning this protection in health and disease remain elusive. Here, we used a Förster Resonance Energy Transfer (FRET)-based cGMP biosensor combined with scanning ion conductance microscopy (SICM) to show that functional β₃-ARs are mostly confined to the T-tubules of healthy rat cardiomyocytes. Heart failure, induced via myocardial infarction, causes a decrease of the cGMP levels generated by these receptors and a change of subcellular cGMP compartmentation. Furthermore, attenuated cGMP signals led to impaired phosphodiesterase two dependent negative cGMP-to-cAMP cross-talk. In conclusion, topographic and functional reorganization of the β₃-AR/cGMP signalosome happens in heart failure and should be considered when designing new therapies acting via this receptor.

## Introduction

Over the last two decades, functional β₃-adrenergic receptors (β₃-ARs) have been found and studied in cardiomyocytes isolated from various species including humans and rodents (*Gauthier et al., 1998*; *Mongillo et al., 2006*; *Hammond and Balligand, 2012*). Depending on the cell type (cardiomyocytes *vs* adipocytes or atrial *vs* ventricular myocytes), β₃-ARs have been reported to couple to both stimulatory ($G_s$) and inhibitory ($G_i$) proteins and to regulate cardiac contractility. In human and rodent ventricular myocardium, catecholamine binding to β₃-ARs elicits negative inotropic and positive lusitropic effects by signalling via $G_i$ and the second messenger 3',5'-cyclic guanosine monophosphate (cGMP) (*Gauthier et al., 1998*; *Mongillo et al., 2006*). Unlike β₁- and β₂-AR, the β₃-AR is resistant to agonist-induced desensitization, (*Liggett et al., 1993*; *Nantel et al., 1993*) and its expression is increased in heart failure as well as in sepsis and diabetic cardiomyopathy (*Amour et al., 2007*; *Moniotte et al., 2007*; *Moniotte et al., 2001*). It was hypothesised that β₃-AR/cGMP pools can attenuate excessive cardiotoxic β₁-AR/cAMP signalling, as well as pathological cardiac hypertrophy and remodelling which takes place in cardiomyocytes during the progression

towards heart failure (*Mongillo et al., 2006*; *Hammond and Balligand, 2012*; *Takimoto et al., 2005*). Endothelial nitric oxide synthase (eNOS), has been detected in close proximity to $\beta_3$-ARs in cardiomyocyte caveolae structures. The caveolae are believed to provide discrete signalling domains, necessary for the autonomic regulation of the heart (*Feron et al., 1998*). It has been shown indirectly that $\beta_3$-AR/cGMP is most likely degraded by the phosphodiesterases 2 and 5 (*Mongillo et al., 2006*; *Takimoto et al., 2005*). Recently, overexpression of $\beta_3$-AR in transgenic mice has been shown to protect the heart from catecholamine-induced hypertrophy and remodelling via an eNOS/soluble guanylyl cyclase (sGC)/cGMP-dependent signalling pathway. The same study showed localization of $\beta_3$-ARs together with eNOS in caveolae-enriched membrane fractions, which had been separated via ultracentrifugation (*Belge et al., 2014*). Another mouse study identified the sGC subunit $\alpha$1 as the facilitator of the NO dependent but $Ca^{2+}$ independent effects of $\beta_3$-AR using sGC $\alpha$1 KO mice (*Cawley et al., 2011*). Despite its name the 'soluble' sGC has been shown to act in close association with $\beta_3$-ARs and membrane located signalosomes (*Mongillo et al., 2006*; *Feron et al., 1998*). However, the exact localization of functional $\beta_3$-ARs in adult cardiomyocytes and the spatio-temporal regulation of their cGMP signals as well as their potential interaction with cAMP signalling pathways have not been studied before.

In this study, we employ a highly sensitive Förster Resonance Energy Transfer (FRET)-based biosensor, red cGES-DE5, in combination with scanning ion conductance microscopy (SICM). We demonstrate that in healthy rat cardiomyocytes, functional $\beta_3$-ARs are localized exclusively within the transverse (T)-tubules and stimulate a cGMP pool which is predominantly regulated by phosphodiesterases (PDEs) 2 and 5. Furthermore, by using the cAMP specific FRET-based biosensor Epac1-camps we show that $\beta_3$-AR stimulation can decrease overall adenylate cyclase dependent cAMP levels in healthy cardiomyocytes by a PDE2-mediated cGMP-to-cAMP cross-talk. This cross-talk appears to be disrupted in heart failure, where $\beta_3$-AR stimulation no longer has a significant effect on overall cAMP levels. In failing cells, $\beta_3$-AR/cGMP signals decrease in the T-tubules. Heart failure leads to altered co-localization of sGC and caveolin-3, as shown via immunocytochemical staining. Together, these alterations result in the impairment of the $\beta_3$-AR-dependent cGMP signalling pathway and of a PDE2-mediated $\beta_3$-AR induced decrease of local cAMP.

## Results

### Echocardiography and biometric data show heart failure phenotype

To study $\beta_3$-AR-dependent cGMP dynamics, we used ventricular cardiomyocytes isolated from healthy and failing rat hearts transduced with an adenovirus to express a highly sensitive cytosolic FRET biosensor red cGES-DE5. As a model of heart failure, we used rats which underwent left coronary artery ligation for 16 weeks (*Lyon et al., 2009*). Echocardiographic and biometric data from these animals are summarized in *Figure 1*. Data analysis showed typical clinical signs of heart failure, including a loss of pump function, left ventricular dilation and left ventricular wall thickening.

### Isoproterenol (ISO) induces a $\beta_3$-AR-dependent cGMP increase in adult rat cardiomyocytes

$\beta$-adrenergic stimulation (ISO, 100 nmol/L) of healthy control rat ventricular cardiomyocytes expressing the cGMP biosensor red cGES-DE5, led to the production of substantial amounts of cGMP (*Figure 2A*) in about 2/3 of all tested cells. In failing cardiomyocytes isolated from rats 16 weeks post-myocardial infarction, administration of the same saturating concentrations of ISO resulted in a significant two-fold reduction in the amount of detectable cGMP (*Figure 2B*, p=0.0465). Blocking $\beta_1$- and $\beta_2$-ARs (with 100 nmol/L CGP20712A and 50 nmol/L ICI118551, respectively) in control cells did not abolish this cGMP production (*Figure 2C*). The signal was however strongly and significantly inhibited in control cells by the application of the $\beta_3$-AR antagonist SR59230A (*Figure 2D*, p=0.0316) or by the nitric oxide synthase (NOS) blocker, nitro-L-arginine methyl ester (L-NAME, *Figure 2E*, p=0.0217).

### $\beta_3$-AR/cGMP is preferentially controlled by PDE2 and PDE5

Next, we stimulated cells with ISO and then applied selective inhibitors of the various cGMP-degrading PDEs to investigate the regulation of $\beta_3$-AR/cGMP dynamics. Following the application of

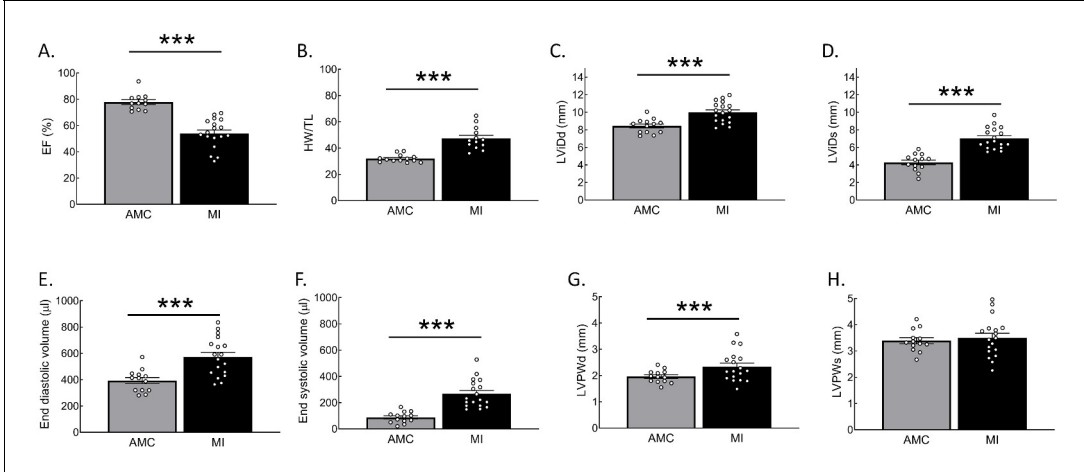

**Figure 1.** Histograms of echocardiography and biometric data in rat age matched control (AMC) hearts and hearts with myocardial infarction (MI). (**A**) Ejection Fraction, (**B**) Heart weight (HW) corrected to tibia length (TL), (**C**) left ventricular diastolic internal dimension (LViDd), (**D**) left ventricular systolic internal dimension (LViDs), (**E**) end-diastolic volume, (**F**) end-systolic volume, (**G**) end-diastolic left ventricular posterior wall thickness (LVPWd), (**H**) end-systolic left ventricular posterior wall thickness (LVPWs). Statistical significance was analysed via two-sided T-test. ***$p<0.001$.

The online version of this article includes the following source data for figure 1:

**Source data 1.** Echocardiography and biometric data for respective treatment groups.

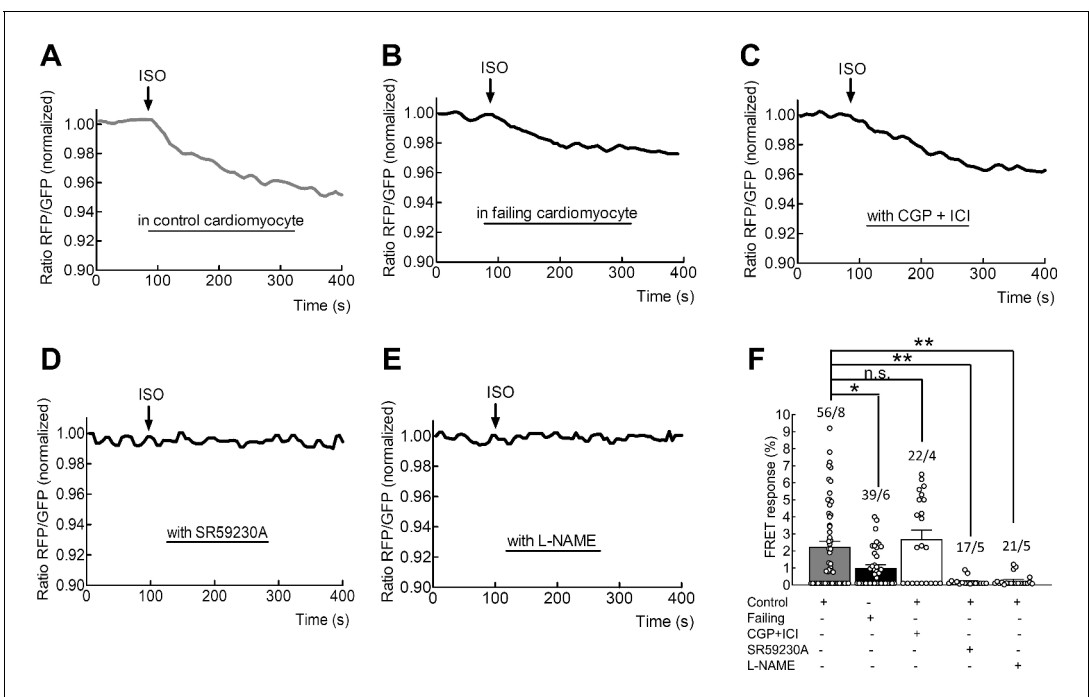

**Figure 2.** Measurements of $\beta_3$-AR-dependent cGMP responses in adult cardiomyocytes. Representative FRET tracings of a control (**A**) or a failing cardiomyocyte (**B**) treated with isoproterenol (100 nmol/L). FRET responses of control cardiomyocytes pre-treated for 5 min with either with the $\beta_1$-AR and $\beta_2$-AR inhibitors CGP20712A (100 nmol/L) and ICI118,551 (50 nmol/L) (**C**), $\beta_3$-AR inhibitor SR59230A (100 nmol/L) (**D**) or for 10 min with the nitric oxide synthase blocker L-NAME (300 μmol/L) (**E**), before the application of isoproterenol (100 nmol/L). (**F**) Quantification of whole cell cGMP-FRET responses from protocols described in A-E). Error bars represent standard error of the mean. Numbers of cells/hearts are shown above the bars. Statistical significance was calculated via Mann Whitney U-test for independent treatments versus control followed by Bonferroni correction: *$p<0.05$; **$p<0.01$.

The online version of this article includes the following source data for figure 2:

**Source data 1.** FRET microscopy data - 'whole-cell' analysis.

selective PDE blockers we applied the non-selective PDE inhibitor IBMX. We found that $\beta_3$-AR/cGMP levels are under the control of multiple PDEs. PDE1 inhibition has a minimal effect on $\beta_3$-AR/cGMP production, whereas PDE2 and PDE5 represent the most prominent $\beta_3$-AR/cGMP degrading families (*Figure 3B,D*). Both PDE2 and PDE5 contribute to more than a half of the overall PDE inhibitory response as determined by IBMX treatment (*Figure 3B,D and E*). Furthermore, we observed that in failing cells, the PDE2, PDE3 and PDE5 inhibitor effects, while not statistically significant, showed a tendency towards increasing (*Figure 3B,D and E*).

## Functional $\beta_3$-ARs are localized in the T-tubules of healthy cells and migrate to the non-tubular sarcolemma in heart failure

Using SICM/FRET we were able to localize functional $\beta_3$-ARs by measuring cGMP-FRET signals following local ligand application from the SICM nanopipette. This approach stimulates cardiomyocytes specifically within T-tubules or on the non-tubular sarcolemma. In healthy cardiomyocytes, we observed that functional $\beta_3$-ARs reside mainly in the T-tubules with very few responses being detectable outside of T-tubules (*Figure 4A–E*), whereas in failing cells, the $\beta_3$-ARs responses after localized stimulation can be detected in both tubulated and non-tubulated areas across the sarcolemma (*Figure 4F–J*).

The increased activity of $\beta_3$-ARs in non-tubulated surface areas in failing cells might be linked to a disrupted association of $\beta_3$-AR with caveolar signalosomes. We investigated this hypothesis by using the cell-permeable peptide disruptor of caveolar signalling TAT-C3SD. The addition of this peptide leads to the dissociation of caveolar signalosomes by inhibiting signalling which is dependent upon the binding to the caveolin-3 specific scaffolding domain (C3SD) (*Macdougall et al., 2012*). In control cells, $\beta_3$-AR-cGMP responses in the T-tubules are higher than in the non-tubulated cell surface areas, as is seen in *Figure 4E*. However, the T-tubular localization of the receptor can be abolished by treating cells with the TAT-C3SD peptide (*Figure 4—figure supplement 1*), so that the response level in the crest areas equals the response level of the T-tubules.

## Heart failure disrupts $\beta_3$-AR associated sGC localization in caveolin-rich membrane domains

To precisely investigate the localization of the components of the $\beta_3$-AR signalosome in control cells and failing cardiomyocytes, we performed immunocytochemical staining of sGC subunits together with caveolin-3 or $\alpha$-actinin. We detected partial co-localization of sGC$\alpha$1 with caveolin-3 (*Figure 5A,C,E*) which was significantly decreased by about 20% in failing cardiomyocytes (*Figure 5E*, p=0.0014). Concomitantly, sGC$\alpha$1 co-localization with the microfilament protein $\alpha$-actinin (*Figure 5B,D,F*) had a tendency to increase in failing cardiomyocytes (*Figure 5F*, p=0.0244), whereas sGC$\beta$1 subunit co-localization with caveolin-3 also decreased by about 16% (*Figure 5—figure supplement 1*, p=0.0044), indicating a redistribution of sGC away from caveolin-rich microdomains.

## Heart failure impairs PDE2 mediated cGMP-to-cAMP cross-talk after $\beta_3$-AR stimulation

To investigate whether the $\beta_3$-AR signals we detected in the experiments above were able to influence cAMP signalling in cardiomyocytes, we expressed the cAMP biosensor Epac1-camps in healthy and failing cardiomyocytes. To analyse the cGMP-to-cAMP cross-talk, we treated the cell with the adenylyl cyclase activator forskolin with and without $\beta_3$-AR agonist CL316,243 (*Figure 6A,B*). In healthy cells, stimulation of $\beta_3$-AR led to a significant reduction of approximately 10.3% of the forskolin stimulated cAMP production (*Figure 6C*, p=0.040). The PDE2 inhibitor BAY60-7550 used in this setting was able to abolish the observed $\beta_3$-AR agonist effects on cAMP levels (*Figure 6A,C*, p=0.0458). We used forskolin to directly activate adenylate cyclases downstream of $\beta$-ARs to allow us a direct comparison between control and MI cells, which might have been complicated otherwise due to $\beta_1$-AR receptor desensitisation. Nonetheless, our result suggests that $\beta_3$-ARs acts via increased PDE2 activation to attenuate cAMP responses. In failing myocytes, the effect of CL316,243 on the cAMP response was no longer significant (*Figure 6B,C*), suggesting that the PDE2 mediated cGMP-to-cAMP crosstalk is disrupted by disease.

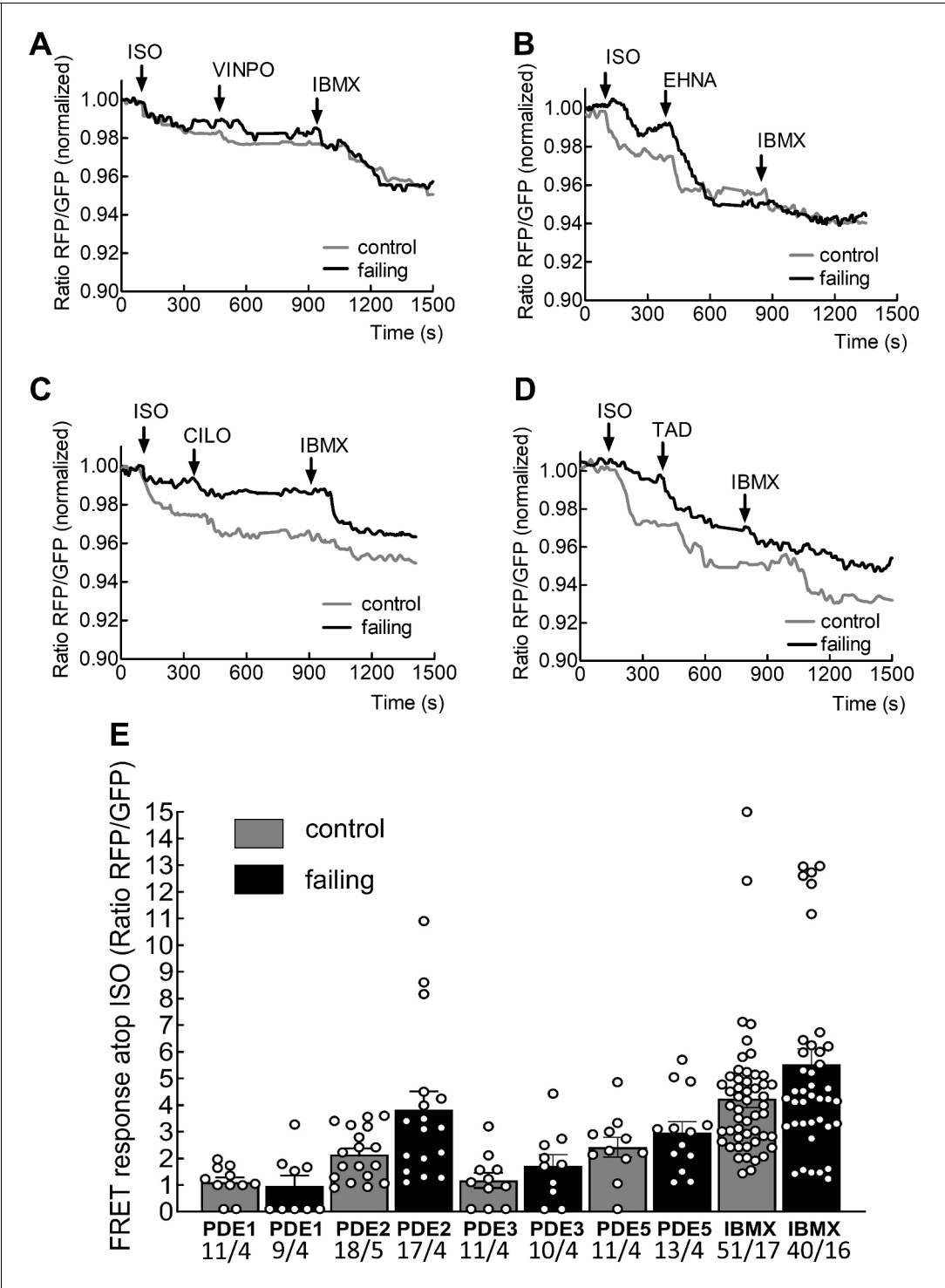

**Figure 3.** Investigation of phosphodiesterase regulation of $\beta_3$-AR/cGMP in adult cardiomyocytes. Representative FRET response curves of control (grey line) and failing (black line) cardiomyocytes following whole cell treatment with isoproterenol (100 nmol/L) followed by the PDE1 blocker vinpocetine (VINPO, 10 μmol/L) (**A**), the PDE2 inhibitor EHNA (10 μmol/L) (**B**), the PDE3 inhibitor cilostamide (CILO, 10 μmol/L) (**C**) and the PDE5 inhibitor tadalafil (TAD, 100 nmol /L) (**D**) followed by the non-specific PDE blocker IBMX (100 μmol/L). The scatter plot/histograms present whole cell cGMP-FRET responses evoked by PDE inhibition further to the isoproterenol responses in % from (**A–D**) (**E**) Error bars represent standard error of the mean. Numbers of cells/hearts are shown below the bars. Statistical significance was calculated via mixed ANOVA followed by χ2-test: No statistically significant differences between control and failing conditions for any PDE could be detected, only tendencies to increased responses for PDE2, PDE3 and PDE5 inhibitors.

*Figure 3 continued on next page*

*Figure 3 continued*

The online version of this article includes the following source data for figure 3:

**Source data 1.** FRET microscopy data - 'whole cell' analysis PDEs.

## Discussion

The pharmacological modulation of $\beta_3$-AR for the treatment of cardiac hypertrophy and heart failure has recently emerged as a promising therapeutic route in translational research (*Belge et al., 2014*). Previous studies have suggested that the $\beta_3$-AR are localized in caveolae, in close proximity to its signalling partners eNOS and sGC (*Mongillo et al., 2006*; *Belge et al., 2014*). However, the exact sub-membrane localization of $\beta_3$-AR and the compartmentation of its signalling to cGMP were not well understood. Alterations in $\beta_3$-AR signalling in disease states have been difficult to study due to the lack of appropriate imaging techniques and specific antibodies which work in situations of relatively low endogenous expression. In this work, we studied the exact submembrane localization of $\beta_3$-AR and alterations to $\beta_3$-AR/cGMP signalling in failing cells using new cutting-edge biophysical approaches such as SICM and FRET. We were able to directly visualize compartmentalized $\beta_3$-AR/cGMP production in the cellular context of adult rat cardiomyocytes and its disruption in heart failure. Using FRET imaging in the presence of either $\beta_3$-AR agonists or antagonists, we show that these cells can produce cGMP following direct agonist stimulation of $\beta_3$-ARs (see *Figure 2*). In some, but not all cells, residual cGMP production was still detectable despite $\beta_3$-AR and NOS inhibition. This is an observation which has previouslybeen reported in neonatal rat cardiomyocytes (*Mongillo et al., 2006*). $\beta_3$-AR dependent cGMP pools are mainly formed in the T-tubules (see *Figure 4C*) due to the localization of $\beta_3$-ARs in close proximity to the caveolar signalosome, which among other molecules is comprised of eNOS and sGC. To further substantiate the association of functional $\beta_3$-AR to caveolae, we dissociated the signalosome of these lipid structures in healthy cardiomyocytes using a previously published peptide, which targets the caveolin-3 scaffolding domain (*Macdougall et al., 2012*). As a result, $\beta_3$-AR-cGMP pools were induced outside of T-tubular domains (see *Figure 4—figure supplement 1*) which corroborates the importance of caveolar signalosomes for proper $\beta_3$-AR regulation.

Using various family selective PDE inhibitors, we uncovered that $\beta_3$-AR/cGMP signals are predominantly regulated by local pools of PDE2 and PDE5 (see *Figure 3B,D and E*). These findings are similar to what has been described for atrial natriuretic peptide-stimulated pools of submembrane cGMP in rat cardiomyocytes (*Castro et al., 2006*) and eNOS/sGC-dependent pools of cGMP in mouse cells (*Takimoto et al., 2005*). Further regulation via the cGMP specific and IBMX insensitive PDE9 cannot be completely excluded, although a recent study in mice suggests that PDE9 degrades cGMP pools generated downstream of natriuretic peptide receptors acting independently of NO (*Lee et al., 2015*). Interestingly, the overall $\beta_3$-AR/cGMP response was significantly reduced in failing cells (*Figure 2B,F*) despite increased $\beta_3$-AR expression reported in cardiac disease (*Hammond and Balligand, 2012*; *Moniotte et al., 2001*). This reduction in the signal could be in part be due to higher PDE activity on cGMP (*Figure 3B,E*) and an altered subcellular arrangement of sGC (*Figure 5*). sGC was found to redistribute away from caveolin-3 in heart failure, as demonstrated in this work by immunocytochemical double staining. We have observed a trend to an increased overlap between our sGC$\alpha$ and $\alpha$-actinin staining in our confocal imaging, which could potentially represent an increased redistribution of sGC$\alpha$ to the areas of the Z-disc not directly associated with the T-tubules or caveolar signalosomes. Though the immunocytochemical method is limited in its spatial resolution and can therefore not resolve the caveolae structures themselves, it allows us to detect an alteration in sGC localization in heart failure, which could potentially be indicative of dysregulated caveolar signalosomes as reported previously in a pressure overload induced heart failure model using mice (*Tsai et al., 2012*). Decreased expression of eNOS in the caveolae, alongside increased expression of neuronal NOS (nNOS) at the sarcoplasmic reticulum has been reported in studies of tissue from humans with heart failure (*Damy et al., 2004*; *Drexler et al., 1998*). As eNOS function in cardiomyocytes has been shown to attenuate beta-adrenergic contractile responses (*Farah et al., 2018*), potential changes in local NOS activities for example due to decreased eNOS-caveolin-3 association (*Feron et al., 1998*) could further contribute to altered $\beta_3$-AR/cGMP signal disruption. The slightly increased contributions of PDE2,3 and 5 to cGMP regulation which we observed in heart failure, are

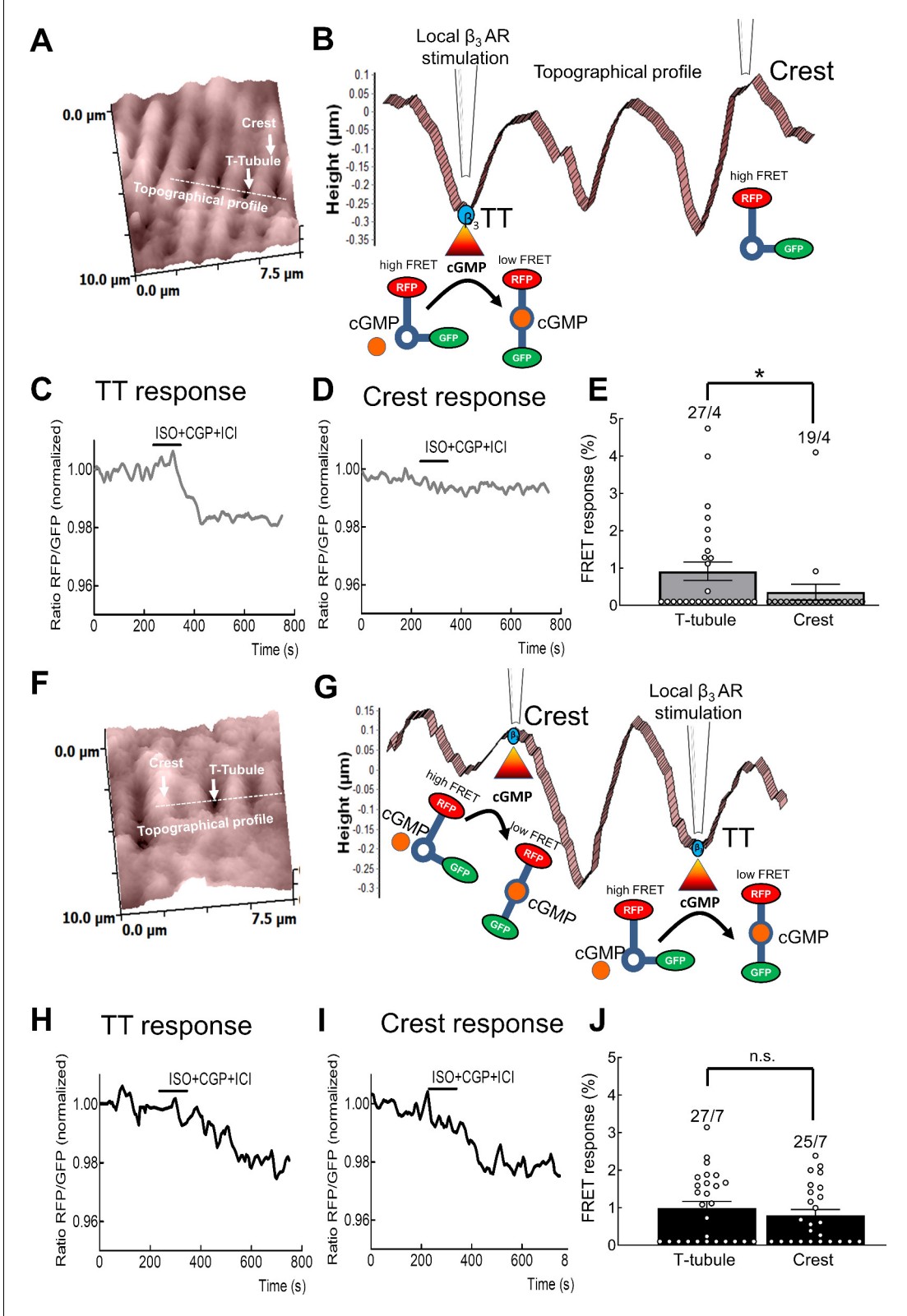

**Figure 4.** Identification of β₃-AR/cGMP signal localization using scanning ion conductance microscopy (SICM) combined with Förster Resonance Energy Transfer (FRET). Representative SICM surface scan of a 10 × 10 µm area of a healthy (**A**) and a failing cardiomyocyte (**F**). White arrows indicate T-tubule or crest structures and a dotted white line indicates the areas selected for the topographical profiles presented in (**B**) and (**G**). Representative topographical profiles of healthy (**B**) and failing cardiomyocytes (**G**). Images present schematics of local β₃-AR stimulation with Isoproterenol (50 µmol/L)

*Figure 4 continued on next page*

*Figure 4 continued*

either inside a T-tubule opening or on the cell crest via the SICM nanopipette. Representative FRET response curves during perfusion with the $\beta_1$-AR blocker CGP20712A (100 nmol/L) and the $\beta_2$-AR blocker ICI118551 (50 nmol/L) and local stimulation inside the T-tubule (C) and the crest of a control cardiomyocyte (D) or the T-tubule (H) and crest (I) of a failing cardiomyocyte. Scatter plots presenting the localised cGMP-FRET responses of control (E) and failing cardiomyocytes (J). Error bars represent standard error of the mean. Numbers of cells/hearts are shown above the bars. *p<0.05, n.s. – not significant by Mann-Whitney U test.

The online version of this article includes the following source data and figure supplement(s) for figure 4:

**Source data 1.** FRET microscopy data - SICM/FRET.
**Figure supplement 1.** The effect of Caveolin-3 displacement on β3AR/cGMP localization measured by SICM/FRET.
**Figure supplement 1—source data 1.** SICM/FRET data TAT-C3SD Treatment.

probably responsible for the overall reduction in cGMP levels, as can be seen when comparing total ISO plus IBMX responses in control and failing cells (*Figures 2F* and *3E*). Increased expression and activity of the dual-specificity PDE2 has been shown in human heart failure (*Mehel et al., 2013*), as well as in a rodent model of aortic banding, where an increase in both cGMP and cAMP hydrolysis took place (*Yanaka et al., 2003*). At the same time as being degraded by PDE2, cGMP can also increase the effect of PDE2 on cAMP by binding to the PDE's GAF-B domain and leading to a conformational change, thereby triggering the so-called cGMP-to-cAMP crosstalk. This crosstalk between the second messengers of which one (cAMP) has increasingly been associated with detrimental signalling pathways in the context of heart failure, which the other (cGMP) could potentially attenuate (*Mongillo et al., 2006*; *Moniotte et al., 2001*; *Yanaka et al., 2003*). Interestingly, we have also detected a decreased $\beta_3$-AR induced attenuation of the forskolin induced cAMP response, due to PDE2 activity, which it was possible to abolished by blocking the PDE2 using a specific inhibitor (see *Figures 6* and *7*). This observed impairment of the cGMP/cAMP crosstalk further supports the hypothesis of dramatic spatial rearrangements in the $\beta_3$-AR/sGC/PDE2 signalosome which impair cGMP dynamics and could potentially lead to depressed $\beta_3$-AR effects on cardiac contractility demonstrated in similar animal models (*Mongillo et al., 2006*) and in human heart tissue samples (*Moniotte et al., 2001*). The observed response of $\beta_3$-AR stimulation of about 10 percent on overall cAMP levels, as measured via the cytosolic FRET sensor Epac1-camps, could be of physiological relevance when brought into the context of lowering pathological cAMP signalling levels on a whole or in confined signalling compartments. We believe that altered compartmentation of subcellular cGMP may play a role in disease-associated changes in β-AR signalling. However, heart failure does not completely ablate $\beta_3$-AR/cGMP responses, leaving room for a residual cardioprotective action of $\beta_3$-ARs in the failing heart. This therapeutic potential is currently being addressed pharmacologically using the $\beta_3$-AR agonist mirabegron in clinical studies to treat heart failure with preserved and reduced ejection fraction (*Pouleur et al., 2018*).

In summary, our study reveals mechanisms of submembrane localization of cardiomyocyte $\beta_3$-AR, which regulates the compartmentation of receptor coupling to cGMP production and disease-driven alterations in $\beta_3$-AR/cGMP signalling. These data add insights to the growing body of data regarding the therapeutic implications for the potential treatment of heart failure by $\beta_3$-AR agonists.

## Materials and methods

### Experimental reagents

M199 medium (Invitrogen UK, 11150), taurine (Biochemica, A1141), creatine monohydrate (Sigma Aldrich, C3630), penicillin/streptomycin (Merck, A2212), carnitine hydrochloride (Sigma Aldrich, C9500) BSA (Sigma Aldrich, A6003), laminin (Sigma Aldrich L2020), isoproterenol hydrochloride (Sigma Aldrich, I6504), ICI118551 (Tocris UK, 0821), CGP 20712A (Tocris UK, 1024), SR 59230A hydrochloride (Tocris UK, 1511), L-NAME hydrochloride (Tocris UK, 0665), Vinpocetine (Sigma Aldrich, V6383), EHNA hydrochloride (Sigma Aldrich, E114), Cilostamide (Tocris UK, 0915), Tadalafil (Santa Cruz USA, sc-208412), IBMX (Santa Cruz sc-201188), self-made rabbit sGCα and β subunit antibodies (specificity tested in KO animals; *Friebe et al., 2018*), mouse α-actinin (Sigma Aldrich, A7732), mouse Caveolin-3 (BD Transduction Laboratories, 610421, specificity tested in KO animals; *Woodman et al., 2002*), secondary Alexa Fluor antibodies 488 nm, 514 nm, 568 nm and 633 nm

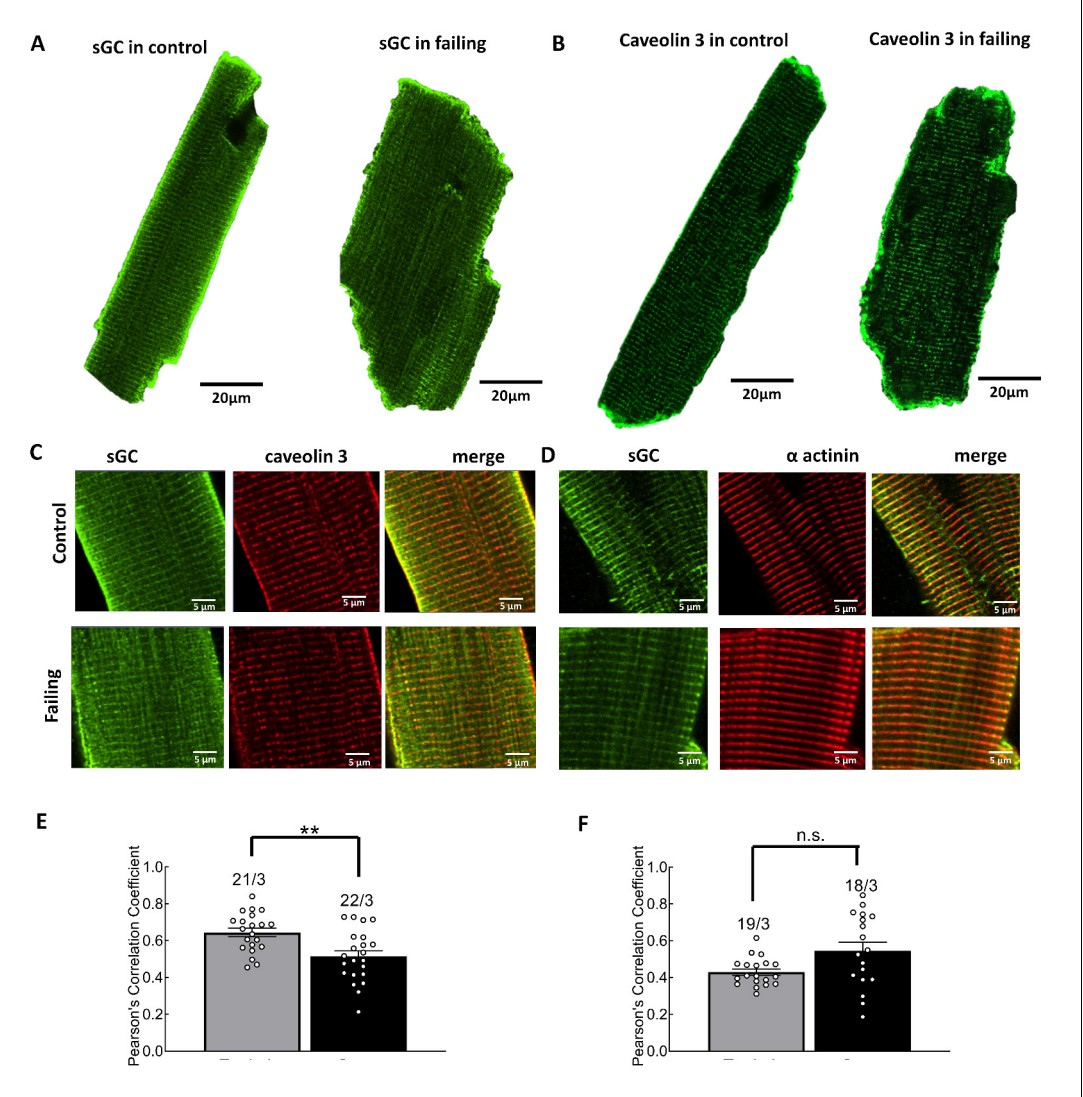

**Figure 5.** Investigation of sGC and caveolin-3 localization in control and failing cardiomyocytes. Representative, confocal images of sGCα1 (A) and caveolin-3(B) in control and failing cardiomyocytes. Magnified representations of double staining of sGCα1(C) with caveolin-3 and of sGCα1 (D) with α-actinin. Quantification of sGC (E) and caveolin-3 and of sGC (F) and α-actinin co-localization in control and failing cells. Error bars represent standard error of the mean. Numbers of cells/hearts are shown above the bars. Statistical significance was analyzed via mixed ANOVA followed by χ2-test; **p<0.01, n.s. – not significant.

The online version of this article includes the following source data and figure supplement(s) for figure 5:

**Source data 1.** Immunostaining data sGC and Cav3 colocalization control.

**Figure supplement 1.** Investigation of sGC and caveolin-3 localization in control and failing cardiomyocytes.

**Figure supplement 1—source data 1.** Immunostaining data sGC and Cav3 colocalization failing cells.

(Life Technologies), BSA (Fisher Scientific UK, BPE9704), fluorescence mounting medium (Vectashield Germany, H-1000), MaTek glass-bottom dishes (MaTek USA, P35G-1.5–10 C), TAT-scram and TAT-Cav3 peptides (a gift from Dr. Sarah Calaghan from Leeds, England).

## Myocardial infarction (MI) model

All animal experiments were performed in the United Kingdom (UK) according to the standards for the care and use of animal subjects determined by the UK Home Office (ASPA1986 Amendments Regulations 2012) incorporating the EU directive 2010/63/EU. The Animal Welfare and Ethical Review Body Committee of Imperial College London approved all protocols.

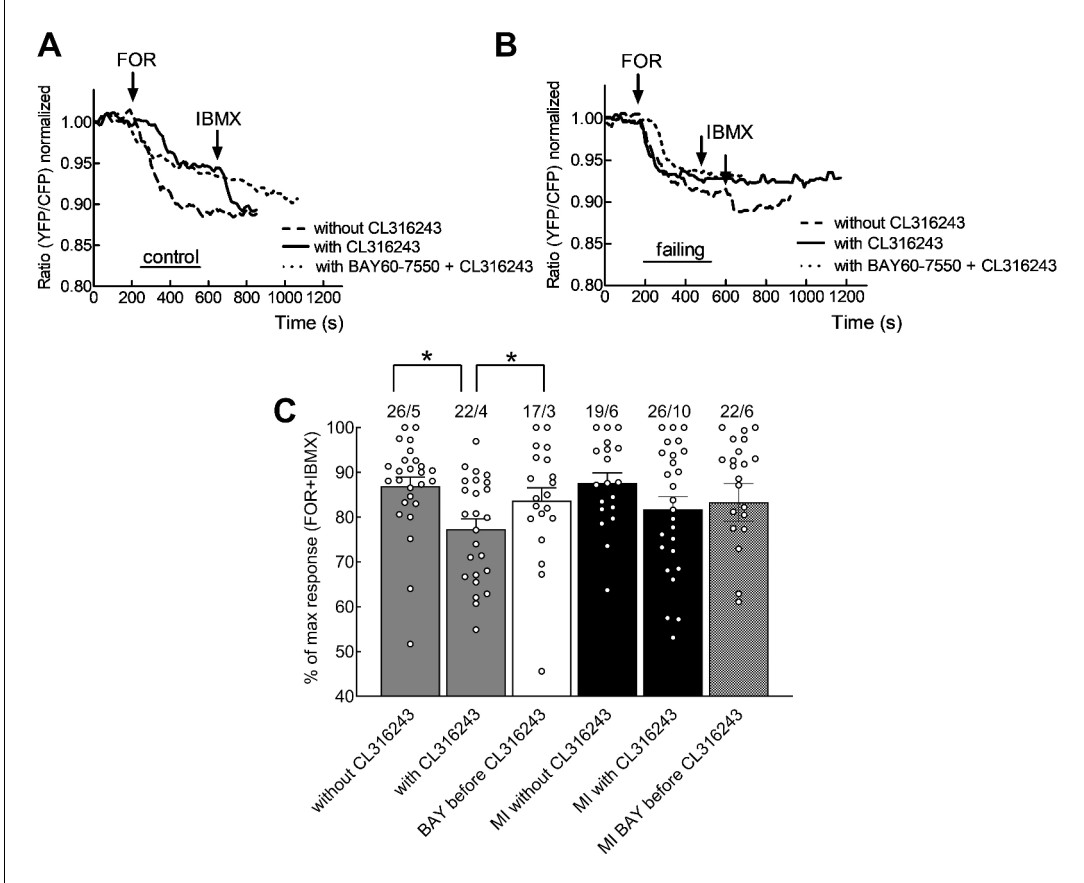

**Figure 6.** β₃-AR signalling can affect cAMP levels via PDE2. Representative FRET responses for control cardiomyocytes (**A**) or failing cardiomyocytes from myocardial infarction (MI) hearts (**B**) stimulated using forskolin (FOR, 10 μmol/L) applied with or without the β₃-AR agonist CL316243 (1 μmol/L) and with or without pre-treatment with the PDE2 inhibitor BAY60-7550 (100 nmol/L). (**C**) Scatter plot/histogram presenting whole cell cAMP-FRET responses depicted as the percentage of the maximal possible cAMP FRET response (=Forskolin followed by IBMX). The measured Forskolin or IBMX responses were the respective maximal responses, equalling the lowest FRET ratio value, achieved after each stimulus. The effect of the β₃-AR agonist on forskolin induced cAMP levels is no longer discernible after inhibition of PDE2 in control cardiomyocytes. This effect is no longer significant in failing cardiomyocytes. Error bars represent standard error mean. Statistical significance was calculated via mixed ANOVA followed by χ2-test* p<0.05. The online version of this article includes the following source data for figure 6:

**Source data 1.** FRET microscopy data cAMP/cGMP crosstalk experiment.

The parts of the investigation (on isolated cardiomyocytes from healthy rats) which were performed in Germany, conformed to the guide for the care and use of laboratory animals published by the National Institutes of Health (Bethesda, Maryland; Publication No. 85–23, revised 2011, published by National Research Council, Washington, DC). The experimental procedures were in accordance with the German Law for the Protection of Animals and with the guidelines of the European Community.

The following procedure was exclusively performed at Imperial College London in the UK: Left descending coronary artery ligation was performed as described (*Nikolaev et al., 2010*). Rats were monitored by echocardiography in M-mode under anaesthesia (2% isoflurane). Animals with induced MI were sacrificed 16 weeks after MI for ventricular cardiomyocyte isolation via enzyme digestion of the Langendorff perfused heart as described (*Lyon et al., 2009*). Age matched animals served as control. Echocardiographic and biometric data are summarized in *Figure 1*. All animals were male.

## Peptide dissociation of caveolae signalosome

For disruption of the caveolar signalosome, cardiomyocytes were treated with a cell-membrane penetrating TAT peptide targeting the Caveolin-3 scaffolding domain (C3SD) (sequence: YGRKK RRQRRRGGGGDGVWRVSYTTFTVSKYWCYR) or with a scrambled control peptide without any

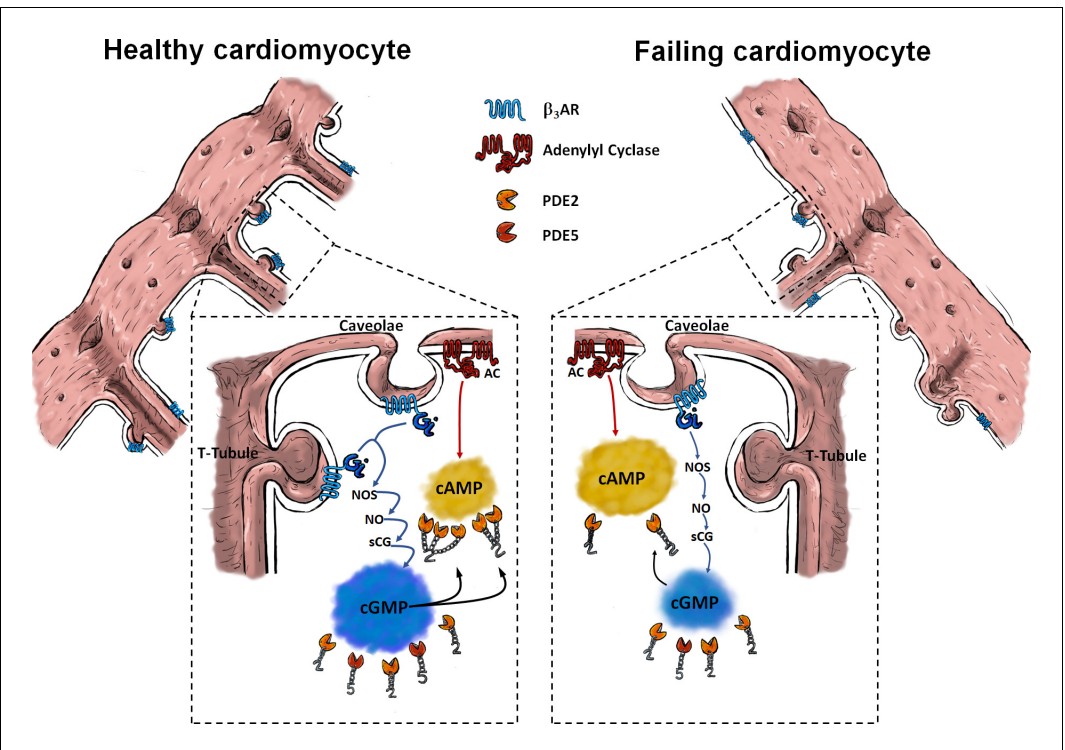

**Figure 7.** Schematic of β₃-AR/cGMP signalling in healthy (left side) and failing (right side) cardiomyocytes. In healthy cardiomyocytes, functional β₃-ARs are associated with caveolar signalosomes and localized mostly in T-tubules. Via Gi/NOS/NO/sGC/cGMP signalling they can suppress strong cAMP responses by stimulating increased PDE2 dependent cAMP degradation through cGMP binding to GAF-B domain of PDE2. In heart failure, increased presence of β₃-AR activity at non-tubular plasma membrane (Crest) and away from caveolin-3 associated membrane domains might disrupt receptor-associated cGMP signalosomes and lead to disrupted cGMP/cAMP-crosstalk.

cellular targets in cardiomyocytes (sequence: YGRKKRRQRRRGGGGYWTVYTKVDFCGSRYVRTSW) as described previously (*Macdougall et al., 2012*). Cells were incubated with peptides by directly putting the peptides into the cell medium for 30 min at 37°C.

### Drug concentrations

The concentrations for pharmacological agonists and antagonists for the β-AR subtypes and the PDE subtypes investigated were taken from previously published work, which established the drug affinity through competitive radioligand binding assays, FRET sensor dose-response levels and PDE activity assays (*Nikolaev et al., 2010*; *Nikolaev et al., 2006*; *Hoffmann et al., 2004*; *Johnson et al., 2012*).

### Whole cell and SICM/FRET cGMP measurements

After isolation, adult rat ventricular myocytes were plated onto laminin-coated cover glasses or Mat-Tek dishes and cultured in M199 media supplemented with creatine 5 mM, taurine 5 mM, carnitine 5 mM, bovine serum albumin 1%, ascorbate 1 mM and penicillin/streptomycin 1%, before being subjected to FRET and SICM/FRET measurements at room temperature as described (*Nikolaev et al., 2010*) 44–52 hr after transduction with adenovirus expressing the cGMP-FRET biosensor red cGES-DE5 (*Götz et al., 2014*) at a multiplicity of infection equal 300. The respective buffer, at pH = 7.4, in which the cells were imaged contained 144 mmol/L NaCl, 5.4 mmol/L KCl, 1 mmol/L MgCl₂, 1 mmol/L CaCl₂ and 10 mmol/L HEPES. Whole cell β₃-AR/cGMP levels were measured by treating control and failing cardiomyocytes with 100 nmol/L isoproterenol. To determine the source of isoproterenol induced cGMP whole cell FRET measurements were performed by pre-blocking control cardiomyocytes either with 100 nmol/L of the β₃-AR inhibitor SR 59230A for 5 min or with 100 nmol/L of the β₁-AR blocker CGP 20712A and 50 nmol/L of the β₂-AR blocker ICI 118,551 for 5 min, or

with 300 µM Nω-Nitro-L-arginine methyl ester hydrochloride for 10 min, before applying 100 nmol/L of isoproterenol. For whole cell FRET measurements of phosphodiesterase dependent regulation of $\beta_3$-AR/cGMP signals, cells were treated with 100 nmol/L isoproterenol followed by either 10 µmol/L of the specific phosphodiesterase blocker vinpocetine for PDE1, 10 µmol/L erythro-9-amino-β-hexyl-α-methyl-9H-purine-9-ethanol (EHNA) for PDE2, 10 µmol/L cilostamide for PDE3 or 100 nmol/L Tadalafil for PDE5, and 100 µmol/L of the unspecific PDE inhibitor 3-Isobutyl-1-methylxanthine (IBMX). To study $\beta_3$-AR/cGMP localization on cardiomyocyte surface structures, cells were continuously superfused with FRET buffer containing 100 nmol/L of the $\beta_1$-AR blocker CGP 20712A and 50 nmol/L of the $\beta_2$-AR blocker ICI 118,551. SICM was used to scan and visualize cardiomyocyte T-tubule openings and crest surface structures. Next, the scanning nanopipette was lowered onto either T-tubule openings or crest structures and receptor ligand was locally applied from the nanopipette filled with 50 µmol/L Isoproterenol (ISO) and 50 µmol/L CGP 20712A and 25 µmol/L ICI 118,551 by switching SICM pipette holding potential from −200 to +500 mV as described (*Schobesberger et al., 2016*).

## FRET-based cAMP measurements

Freshly isolated cardiomyocytes were transduced with Epac1-camps (*Nikolaev et al., 2004*) adenovirus virus for 44–52 hr and exposed to whole cell FRET measurements for $\beta_3$-AR effects on cAMP levels. Cells were kept in FRET buffer containing 50 nmol/L $\beta_2$AR inhibitor ICI 118,551 before treating them with 100 nmol/L of the $\beta_3$-AR agonist CL 316,243 plus 50 nmol/L ICI 118,551 or just adding FRET buffer with 50 nmol/L ICI 118,551 followed by addition of 10 µmol/L Forskolin and consequently 100 µmol/L IBMX into the solution. Additional cells were treated using the same protocol but with or without 100 nmol/L of the PDE2 inhibitor BAY 60–7550 in the cell bath.

## Confocal imaging

Freshly isolated cardiomyocytes were plated onto laminin coated coverslips and fixed in ice cold methanol for 10 min. Next, they were blocked with PBS based buffer containing 0.3% Triton X-100% and 5% fetal calf serum. Cells were then double stained overnight, at 4°C with primary antibodies, diluted in PBS containing 0.3% Triton and 1% BSA, for example against sGC$\alpha$1 together with $\alpha$-actinin or Caveolin-3. Cells were then washed three times for 5 min with PBS before they were exposed to secondary antibodies diluted in PBS containing 0.3% Triton and 1% BSA for 1 hr. Finally, cells were washed again in PBS three times for 5 min before being mounted onto cover slides with mounting medium. Imaging was performed using an inverted Zeiss LSM-800 laser scanning microscope equipped with a 40x oil immersion objective and controlled by ZEN imaging software which was used for the colocalization analysis and detection of the respective Pearson's correlation coefficients.

## Statistics

Statistical differences were analysed using OriginPro 8.6 and GraphPad Prism 7 software. Normal data distribution was determined by Kolmogorov-Smirnov test. For the comparison of two independent groups with skewed distribution Mann-Whitney U test was used. For the comparison of normally distributed data, a two-tailed T-test (for comparing morphometric and echocardiographic data from independent animals, *Figure 1*) or a mixed ANOVA followed by $\chi$2-test (when data from several cells from multiple individual animals were compared) were applied. Differences were considered significant at p-values below 0.05. All data are presented as means ± s.e.m.

## Acknowledgements

We thank Peter O'Gara for cardiomyocyte isolation. Confocal microscopy was performed at the FILM imaging facility of Imperial College London and at the Department of Pharmacology and Toxicology, University Medical Centre Hamburg-Eppendorf.

## Additional information

### Funding

| Funder | Grant reference number | Author |
|---|---|---|
| British Heart Foundation | 12/18/30088 | Julia Gorelik |
| Wellcome | WT090594 | Julia Gorelik |
| Deutsche Forschungsgemeinschaft | Fr 1725/3-2 | Andreas Friebe<br>Viacheslav O Nikolaev |
| National Institutes of Health | ROI-HL grant 126802 | Julia Gorelik |
| Deutsche Forschungsgemeinschaft | NI 1301/3-2 | Andreas Friebe<br>Viacheslav O Nikolaev |

The funders had no role in study design, data collection and interpretation, or the decision to submit the work for publication.

### Author contributions

Sophie Schobesberger, Formal analysis, Investigation; Peter T Wright, Data curation, Formal analysis, Investigation, Methodology; Claire Poulet, Data curation, Investigation, Visualization, Methodology; Jose L Sanchez Alonso Mardones, Formal analysis, Investigation, Methodology; Catherine Mansfield, Resources, Data curation, Methodology; Andreas Friebe, Conceptualization, Resources, Funding acquisition; Sian E Harding, Conceptualization, Resources, Supervision; Jean-Luc Balligand, Conceptualization; Viacheslav O Nikolaev, Julia Gorelik, Conceptualization, Resources, Software, Supervision, Funding acquisition, Methodology

### Author ORCIDs

Sophie Schobesberger (iD) https://orcid.org/0000-0001-8268-0019
Peter T Wright (iD) https://orcid.org/0000-0002-6504-590X
Viacheslav O Nikolaev (iD) https://orcid.org/0000-0002-7529-5179
Julia Gorelik (iD) https://orcid.org/0000-0003-1148-9158

### Ethics

Animal experimentation: All procedures performed in the UK were carried out according to the standards for the care and use of animal subjects determined by the UK Home Office (ASPA1986 Amendments Regulations 2012) incorporating the EU directive 2010/63/EU. The Animal Welfare and Ethical Review Body Committee of Imperial College London approved all protocols. The parts of the investigation, which were performed in Germany, conformed to the guide for the care and use of laboratory animals published by the National Institutes of Health (Bethesda, Maryland; Publication No. 85-23, revised 2011, published by National Research Council, Washington, DC). The experimental procedures were in accordance with the German Law for the Protection of Animals and with the guidelines of the European Community.

### Decision letter and Author response

Decision letter https://doi.org/10.7554/eLife.52221.sa1
Author response https://doi.org/10.7554/eLife.52221.sa2

## Additional files

### Supplementary files

• Transparent reporting form

## Data availability

All data generated or analysed during this study are included in the manuscript and supporting files. Source data files are provided for all figures.

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
