## [Decision Letter]

**Acceptance summary:**

Several complementary biophysical methods are used to explore β_3_-Adrenoceptor localization in normal and impaired cells. The results show that redistribution of the receptor impairs NO/cGMP/PDE2 signaling in failing cardiomyocytes. This is interesting and timely because of therapeutic implications when targeting the receptor.

**Decision letter after peer review:**

Thank you for submitting your article "β_3_-Adrenoceptor redistribution impairs NO/cGMP/PDE2 signaling in failing cardiomyocytes" for consideration by *eLife*. Your article has been reviewed by three peer reviewers, and the evaluation has been overseen by a Reviewing Editor and Anna Akhmanova as the Senior Editor. The following individual involved in review of your submission has agreed to reveal their identity: Marc Freichel (Reviewer #3).

The reviewers have discussed the reviews with one another and the Reviewing Editor has drafted this decision to help you prepare a revised submission.

Summary:

Schobesberger et al. analyze the localization of β_3_-adrenergic receptor and associated signaling molecules in signaling domains of adult rat cardiomyocytes and its redistribution in a model of myocardial ischemia. They use FRET-based cGMP and cAMP sensors and scanning ion conductance (SICM) and confocal microscopy. The authors show Isoproterenol-evoked cGMP generation in control cardiomyocytes, its contribution by β_3_-adrenoceptors using β_1_ and β_2_ antagonists and its dependence on NO generation. The authors can demonstrate that ISO-evoked GMP production is reduced in failing cardiomyocytes. Using a large set of phosphodiesterase inhibitors, they elaborate the contribution of PDE2 and PDE5 isoforms for ISO-evoked cGMP generation. Combined FRET and SICM surface imaging reveals that β_3_-adrenoceptor-mediated cGMP generation occurs predominantly in the T tubular cleft in healthy myocytes, and that in failing myocytes the cGMP response is similar in T tubules and at the crest. Modulation of caveolin binding similarly equalizes T-tubule and crest signalling in healthy myocytes. Using confocal microscopy, the authors then show data suggesting that the β_3_-adrenergic receptor-associated signaling molecules sGCα1 and sGβ1 dissociate from caveolin 3-containing domains in failing cardiomyocytes. In a final set of experiments the authors demonstrate that β_3_-adrenoceptor activation attenuates cAMP generation (also measured by a FRET biosensor) in myocytes from healthy animals but that this control is lost in failing cells.

Opinion:

The study provides some interesting new data and is relevant because unlike the β_1_- and β_2_-ARs, β_3_-ARs are not desensitized in HF, and their activator is currently undergoing a Phase IIb clinical trial for treating heart failure. The combined SICM and FRET methods are unique to this investigative group and are valuable for the question at hand. There are a number of intriguing results presented, and the presentation is generally quite clear. However, the conclusions of the study are somewhat overstated, based on the data presented (and the way these data are analysed). This must be addressed.

Essential revisions:

1) First and foremost, recent work of PC Simpson showed clearly that β_3_-AR does not exist in murine cardiac myocytes. The authors should examine their data in view of these observations or argue against.

2) Evidence that the myocytes are from a state of heart failure should be included in the Results (echo data).

3) The choice of drug concentrations (β_1_-AR, β_2_-AR antagonists, PDE inhibitors) must be justified – selectivity of drugs for the targets is key to the interpretation of most data in this manuscript.

4) There are issues with use of statistical tests. MW or t-tests are used to compare multiple groups (Figure 1, Figure 4—figure supplement 1, Figure 5). This should be re-done with ANOVA or non-parametric equivalent. Please notice also other comments about statistical analysis of the data.

5) Interpretation of t-tubule/crest responses (Figure 3 and Figure 4—figure supplement 1) does not appear justified by the data. In Figure 3, comparison of E and J suggest that T-tubule responses are lost but not that they are redistributed to other areas of the sarcolemma. By contrast, comparison of the impact of the C3SD peptide and its scrambled control in healthy cells suggests a gain of crest signalling, rather than a loss of t-tubular signals.

6) Care must be taken with interpretation of work using the C3SD peptide. The sequence replicates the so-called caveolin scaffolding domain (CSD) which for many years was considered to mediate the regulatory role of caveolins through interaction with a complementary caveolin binding motif (CBM) in target proteins. This central tenet has been questioned within the last decade by work from the Parton and Dart laboratories which show that most CBM in caveolin's 'binding partners' are inaccessible for interaction with the CSD. This does not necessarily weaken the evidence for a role for CSD, which may exert effects via alternative mechanisms. However, we are not aware of any evidence that the C3SD peptide leads to 'deletion of intact caveolae'. The cartoons (Figure 4—figure supplement 1C, D) imply that Cav3 (presumably the red line, although not defined in legend) is a caveolar neck protein, which is not accurate. Furthermore, these cartoons suggest that with the C3SD peptide, the t-tubular cGMP signal is reduced which is not borne out by the data in part A of this figure. The legend states 'in cardiomyocytes treated with..C3SD..the cGMP signal in…crest areas is even stronger than when stimulated in t-tubules' but the comparison between these 2 datasets is not made (also see 3 above).

7) It is important to acknowledge the limitation of confocal imaging to provide robust support for the location of proteins in caveolae (lateral resolution ≈200 nm, caveolae 50-100 nm). Could more meaningful information be gained from confocal images (Figure 4) by a separation of interior (t-tubular) and surface (crest) staining?

8) The increased co-localisation of sGC and α-actinin is not explained (Figure 4F) – this seems to suggest increased sCG at z discs, i.e., t-tubules in failing cells which is not consistent with the overall interpretation of data.

9) Figure 5 – the magnitude of the impact of β_3_-AR stimulation on cAMP production (indexed with the Epac-based sensor) is small even in control cells. If this significant difference is retained when the data are analysed using ANOVA, the authors should comment on the relevance of such a small change. Perhaps measurement of cAMP signals in different subcellular compartments might reveal profound differences.

10) No direct evidence is presented here that β_3_-AR are caveolae-based (presumably because of lack of an appropriate antibody) yet some unsubstantiated statements are made e.g. 'β_3_ -AR dependent cAMP pools are formed in caveolae'. Some evidence (referenced in the manuscript) supports the location of some sGC subunits in buoyant caveolae-containing fractions, and the evidence for caveolar location of eNOS is robust. However, comprehensive evidence to support statements regarding the location of the β_3_ signalosome in caveolae is lacking.

11) Figure 6. There seems to be some lack of accuracy in depicting current (and past) data from this group. Why are β1-AR only in crest in healthy myocytes (vs. Nikolaev et al. 2010 Science). Are the PDE2/5 symbols misplaced at the bottom of panel A (surely these degrade cGMP in control cells as shown in Figure 2). The data in Figure 5C suggest that the normal β_3_-AR dependent reduction in cAMP seen in control cells is absent in failing cells (i.e. cAMP lower in control cells in the presence of β_3_-AR stimulation). Yet comparison of panel A and B in Figure 6 suggests that cAMP is lower in failing cells in the presence of β_3_-stimulation. Even given an increase in PDE2 expression/activity in failing cells (not shown explicitly) which could increase degradation of cAMP, how can the data in Figure 5 be reconciled with the cartoons in Figure 6?

12) It appears that the differently targeted β_3_-AR stimuli applied to T-tubules vs. Crest membrane regions were performed in different cells, with comparisons between them analyzed with non-paired statistics. This is not explicitly stated but is apparent by the statistical test used (Mann Whitney U). This seems problematic given that the level of FRET response varies from cell to cell, and even accepting IBMX-based normalization between cells, the statistical power to identify quantitative changes would be challenging. If getting both stimuli in the same cell is impossible and unpaired analysis is required, then the ability to properly normalize between cells is critical. The concern with it is raise by the key data shown in Figure 3. While the extent to which β_3_-AR-mediates cGMP at the Crest appears similar between healthy and failing cardiomyocytes, the FRET response at T-tubule declines in the latter group. If this was due to a redistribution of β_3_-AR to the Crest region, would you not expect a rise in the% FRET response there? Is it possible the results can be explained by a decline in β_3_-R levels or signal-coupling in the T-tubules after MI? Data in Figure 3E and 3J should be combined into a 2WANOVA, and a statistical test made for the interaction between heart failure (or not) and location where the agonist was applied.

13) This study is not the first to examine β_3_ signaling in plasma membrane compartments or its modulation by heart failure (see Trappanese et al., Basic Res Cardiol, 2015), however there are differences in their findings to the current ones that deserve comment. They too found β_3_R expressed in both caveolin enriched and heavy (non-lipid raft) regions shown by direct Western Blot. Their results differ however, in that the distribution in each region did not appear to change in the canine HF model (mitral regurgitation), and signaling seemed only observed if the β_3_R was stimulated in the non-lipid raft compartment. The methods were clearly different, but results should be discussed, particularly in light of Comment 12. Ideally, using a membrane fraction, then Cav3 pull down, and β_3_-AR protein analysis would help confirm if there is indeed a redistribution in this model that was not observed in the canine model reported previously. If there is a real difference, the question then becomes what goes on in human – is it more like rat or dog?

14) There are a number of figures shown where the FRET tracings themselves do not appear to represent the summary data. Examples are in Figure 2 and 5. There are also concerns that for some analysis, very few animals were used, (e.g. Figure 1F, Figure 4—figure supplement 1A, B, Figure 5—figure supplement 1, Figure 5C), and this raises concerns about reproducibility.

15) Figure 5 is another key figure in the study, looking at potential crosstalk between β_3_-AR-stimulated cGMP and cAMP via PDE2 regulation. The modest decline from β_3_-AR agonist in controls seems to be reproduced in the post MI cells too, though the statistical variance is slightly different so one is borderline significant and the other not. The proper test would be a 2WANOVA, where the impact of heart condition is one variable, and the β_3_-AR application the other. This could be done for the β_3_-AR response, and then for +/- PDE2-I response where β_3_-AR stimulation is present. To support the authors' conclusions, there would need to be a significant interaction.

The authors choose forskolin to stimulate cAMP and PDE2-dependent hydrolysis, but this should be contrasted to a β-AR agonist to better support their conclusion (e.g. as depicted in Figure 6).

16) The novelty of the immunofluorescent microscopy data is questioned by prior reports showing similar co-distribution of sGC and Caveolin 3 in normal and failing hearts (e.g. Tsai et al., 2012). There is also a concern that the microscopy is based on freshly isolated cells, whereas the functional data were obtained 48 hours post isolation (time needed to introduce the FRET probes). What is the immunohistology distribution after this 48 hours? Myocytes are known to change the localization of ion channels and various GPCRs during this time period.

17) The MI model is not described, so it is difficult to know the severity of dysfunction generated. Equally important, the location in the heart from which myocytes cells were isolated post MI – is not noted. This is important; e.g. was this peri-infarct, or from the remote territory? How was that determined? How reduced was the EF generated and what other evidence was there supporting this as heart failure – e.g. elevated filling pressures, pulmonary edema, etc.. What was the sex of the animals?

18) The scheme shown in Figure 6 is interesting but not really confirmed by the data shown and so remains speculative. This is because Figure 5 does not test the relative signaling involved in the two different compartments (TT vs. Crest), and with cAMP being stimulated coupled to a Cav3/localizing β-AR, versus not.

19) In Figure 1B and f the authors show a reduced cGMP generation after Isoproterenol stimulation. However, it is not clear whether this is entirely mediated via β_3_ receptors. The authors should do the similar experiments in the presence of CGP plus ICI (β_1_ and β_2_ adrenoceptor blockers).

20) In Figure 2E it is not entirely clear whether the FRET response by IBMX (about 4%) represents the cGMP generation on top of Isoproterenol evoked cGMP generation. If not the IBMX effect together with Isoproterenol is not larger than with ISO alone (Figure 1F). This should be clarified.

21) In the first paragraph of the subsection “Functional β_3_-ARs are localized in the T-tubules of healthy cells and redistribute to the non166 tubular sarcolemma in heart failure”, the authors state that in failing hearts cGMP generation can be detected across the sarcolemma. The authors should more specifically state that in healthy cardiomyocytes there is no crest response which becomes now obvious in failing hearts. At least this is demonstrated in Figure 3D and I.

22) The immunostainings shown in Figure 4 and Figure 5—figure supplement 1 entirely depend on the specificity of the antibodies directed against sGCα1, caveolin3 and sGCβ1. The authors should comment in the manuscript about what is known about the specificity of these antibodies. Have they been used previously in knockout control cells or what other efforts have been made to demonstrate their specificity?

23) Figure 4—figure supplement 1: In the model in figure B and C, it is suggested that cGMP formation is reduced in the T tubules after induction of heart failure. However, if there is no statistical significance Figure 4—figure supplement 1A, B between the first bar (T tubule scramble) and the third bar (T tubular C3SD). Thus, this model is not supported entirely by the data.

24) In Figure 5 the authors should somehow indicate more clearly at which time points the changes in cAMP generation are quantified in C. Is this at the time point just before IBMX application? The authors should clearly state in the manuscript the percentage of inhibition in cAMP generation by the β_3_-agonists. It seems to be in the range of 10-15%?

25) The model in Figure 6 in healthy cardiomyocytes indicates that cGMP act on PDE2 to limit cAMP generation. The authors should elaborate in a little bit more detail how this is achieved for readers not entirely familiar with the previous literature.

---

## [Author Response]

Essential revisions:1) First and foremost, recent work of PC Simpson showed clearly that β_3_-AR does not exist in murine cardiac myocytes. The authors should examine their data in view of these observations or argue against.

We are aware of the recent publication from the PC Simpson group describing single cell PCR analysis of all adrenergic receptor types in wildtype mice (Circ Res. 2017; 120(7): 1103-1115). However, in this manuscript we deal with rat myocytes only. Therefore, we can’t comment on the mouse cells.

Different publications have clearly shown that there are species differences in the expression profiles of adrenergic receptors and thatβ_3_-AR are indeed expressed in rat cardiomyocytes. Also, it is proven that they are overexpressed in various disease states in the rats (i.e. The effect of diabetes on expression of β_1_-, β_2_-, and β_3_-adrenoreceptors in rat hearts. Dinçer UD, Bidasee KR, Güner S, Tay A, Ozçelikay AT, Altan VM. Diabetes. 2001 Feb;50(2):455-61; The Trend of β_3_-Adrenergic Receptor in the Development of Septic Myocardial Depression: A Lipopolysaccharide-Induced Rat Septic Shock Model. Yang, N., Shi, X. L., Zhang, B. L., Rong, J., Zhang, T. N., Xu, W., and Liu, C. F., Cardiology, (2018), 139(4), 234–244; Cardiac effects of long-term active immunization with the second extracellular loop of human β_1_- and/or β_3_-adrenoceptors in Lewis rats. Montaudon E, Dubreil L, Lalanne V, Vermot Des Roches M, Toumaniantz G, Fusellier M, Desfontis JC, Martignat L, Mallem MY. Pharmacol Res. (2015), 100:210-9).

Additionally, in many publications the effects of β_3_-AR specific agonists and antagonists have been reported on the rat cardiomyocytes (i.e. Anti-hypertrophic and antioxidant effect of β_3_-adrenergic stimulation in myocytes requires differential neuronal NOS phosphorylation. Watts VL, Sepulveda FM, Cingolani OH, Ho AS, Niu X, Kim R, Miller KL, Vandegaer K, Bedja D, Gabrielson KL, Rameau G, O'Rourke B, Kass DA, Barouch LA. J Mol Cell Cardiol. (2013), 62:8-17.; Nebivolol: a multifaceted antioxidant and cardioprotectant in hypertensive heart disease. Khan MU, Zhao W, Zhao T, Al Darazi F, Ahokas RA, Sun Y, Bhattacharya SK, Gerling IC, Weber KT. J Cardiovasc Pharmacol. (2013), 62(5):445-51).

Moreover, we have already expressed our criticism of the PC Simpson group publication in an open letter, stating that their methodological approach may not represent the presence of β_3_-AR correctly, particularly not in disease states, where they are upregulated: https://doi.org/10.1161/CIRCRESAHA.117.310942).

Nevertheless, in our experiments we also noticed that not every cell responded to Isoproterenol treatment with a distinct cGMP FRET signal, therefore we added “β-adrenergic stimulation (ISO, 100 nmol/L) of healthy control rat ventricular cardiomyocytes expressing the cGMP biosensor red cGES-DE5, led to the production of substantial amounts of cGMP (Figure 1A) in about 2/3 of all tested cells” into our Results. Additionally, we changed the graphs containing Isoproterenol stimulation of cGMP into scatter plots, to make the non-responders visually discernible from the responding cells (see Figures 2, 4 and Figure 4—figure supplement 1).

2) Evidence that the myocytes are from a state of heart failure should be included in the Results (echo data).

We agree with the reviewer and have added the respective echo data as Figure 1.

Additionally, we added the following passage into the Results section:

“Echocardiographic and biometric data show heart failure phenotype. Echocardiographic and biometric data were collected and are summarised in Figure 1.”

And the following sentence into the Materials and methods section: “Echocardiography and biometric data are summarised in Figure 1.”

3) The choice of drug concentrations (β_1_-AR, β_2_-AR antagonists, PDE inhibitors) must be justified – selectivity of drugs for the targets is key to the interpretation of most data in this manuscript.

The choice of drug concentrations in this manuscript is based on our previous work/publications: Nikolaev et al., 2006; Nikolaev et al., 2010; Hoffmann et al., 2004.

The above publications are providing the drug concentrations for the β-Adrenergic subtypes.

Johnson et al., 2012.

We added the following paragraph into the Materials and methods section about the choice of concentrations “Drug concentrations. The concentrations for pharmacological agonists and antagonists for the β-AR subtypes and the PDE subtypes investigated were taken from previously published work, which established the drug affinity through competitive radioligand binding assays, FRET sensor dose-response levels and PDE activity assays.” with references to the above papers.

4) There are issues with use of statistical tests. MW or t-tests are used to compare multiple groups (Figure 1, Figure 4—figure supplement 1, Figure 5). This should be re-done with ANOVA or non-parametric equivalent. Please notice also other comments about statistical analysis of the data.

We thank the reviewer for the observation and have now performed renewed statistical tests in most figures using either non-parametric Mann Whitney U or a mixed ANOVA followed by χ2-test with cells assigned to the respective rats they were isolated from. As a result, Figure 2 and Figure 5E lost their statistical significance, but are retained in the manuscript as Figure 3 shows the main PDEs degrading β_3_-AR-dependent cGMP and both figures show a potential trend of altered cGMP regulation.

5) Interpretation of t-tubule/crest responses (Figure 3 and Figure 4—figure supplement 1) does not appear justified by the data. In Figure 3, comparison of E and J suggest that T-tubule responses are lost but not that they are redistributed to other areas of the sarcolemma. By contrast, comparison of the impact of the C3SD peptide and its scrambled control in healthy cells suggests a gain of crest signalling, rather than a loss of t-tubular signals.

We thank the reviewer for the suggestion and have reworded the text accordingly as follows:

“The increased activity of β_3_-ARs in non-tubulated surface areas in failing cells might be linked to a disrupted association of β_3_-AR with caveolar signalosomes. […] However, this difference can be abolished by treating cells with the TAT-C3SD peptide, but not with a scrambled peptide, with the level the response on the crest areas increasing to the level of T-tubules (Figure. 1, p=0,0458).”

In addition, to better visualise the cell responses to β_3_-AR stimulation we changed the presentation of Figures 1,2 and 4.

6) Care must be taken with interpretation of work using the C3SD peptide. The sequence replicates the so-called caveolin scaffolding domain (CSD) which for many years was considered to mediate the regulatory role of caveolins through interaction with a complementary caveolin binding motif (CBM) in target proteins. This central tenet has been questioned within the last decade by work from the Parton and Dart laboratories which show that most CBM in caveolin's 'binding partners' are inaccessible for interaction with the CSD. This does not necessarily weaken the evidence for a role for CSD, which may exert effects via alternative mechanisms. However, we are not aware of any evidence that the C3SD peptide leads to 'deletion of intact caveolae'.

We thank the reviewer for this correction and have reworded this part in the manuscript to “We investigated this hypothesis by using the cell-permeable peptide disruptor of caveolar signalling TAT-C3SD. The addition of this peptide leads to the dissociation of caveolar signalosomes by inhibiting signalling which is dependent upon the binding to the caveolin-3 specific scaffolding domain (C3SD)(MacDougall et al., 2012)”, while citing the MacDougall et al., 2012 publication from the Calaghan S. group.

The cartoons (Figure 4—figure supplement 1C, D) imply that Cav3 (presumably the red line, although not defined in legend) is a caveolar neck protein, which is not accurate. Furthermore, these cartoons suggest that with the C3SD peptide, the t-tubular cGMP signal is reduced which is not borne out by the data in part A of this figure. The legend states 'in cardiomyocytes treated with..C3SD..the cGMP signal in…crest areas is even stronger than when stimulated in t-tubules' but the comparison between these 2 datasets is not made (also see 3 above).

We thank the reviewer for this observation and have changed the cartoon to portray Cav 3 more accurately. We also performed additional C3SD peptide measurements for the Figure 5—figure supplement 1.

7) It is important to acknowledge the limitation of confocal imaging to provide robust support for the location of proteins in caveolae (lateral resolution ≈200 nm, caveolae 50-100 nm). Could more meaningful information be gained from confocal images (Figure 4) by a separation of interior (t-tubular) and surface (crest) staining?

We thank the reviewer for this observation and the suggestion to look at different cell areas in our confocal images. We have performed further Pearson Correlation Coefficient measurements by drawing the regions of interest so they either measured the membranes or the cell interior. We could however detect no significant differences in the cell areas. We added a remark about the limitation of confocal microscopy to detect structures underneath the diffraction limit in the Discussion, as follows: “Though the immunocytochemical method is limited in its spatial resolution and can therefore not resolve the caveolae structures themselves, it allows us to detect an alteration in sGC localisation in heart failure and, which could potentially be indicative of dysregulated caveolar signalosomes as reported previously in a pressure overload induced heart failure model using mice (Tsai et al., 2012)”.

8) The increased co-localisation of sGC and α-actinin is not explained (Figure 4F) – this seems to suggest increased sCG at z discs, i.e., t-tubules in failing cells which is not consistent with the overall interpretation of data.

We understand the point the reviewer is making, however after more stringent statistical testing Figure 5F has lost its significance. Nonetheless, we opted to leave the images and graph in the manuscript as it might give a potential trend of where the sGCα subunit might be relocating to in heart failure we and have added the following sentence in the Discussion section: “We have observed a trend to an increased overlap between our sGCα and α-actinin staining in our confocal imaging, which could potentially represent an increased redistribution of sGCα to the areas of the Z-disc not directly associated with the T-tubules or caveolar signalosomes.”

9) Figure 5 – the magnitude of the impact of β_3_-AR stimulation on cAMP production (indexed with the Epac-based sensor) is small even in control cells. If this significant difference is retained when the data are analysed using ANOVA, the authors should comment on the relevance of such a small change. Perhaps measurement of cAMP signals in different subcellular compartments might reveal profound differences.

We have acquired new data that and added to the figure, this effect is statistically significant (see Figure 6).

However, we appreciate the reviewer’s comment and have added the following paragraph into the Discussion:

“The observed response of β_3_-AR stimulation of about 10 percent on overall cAMP levels, as measured via the cytosolic FRET sensor Epac1-camps, could be of physiological relevance when brought into the context of lowering pathological cAMP signalling levels on a whole or in confined signalling compartments.”

10) No direct evidence is presented here that β_3_-AR are caveolae-based (presumably because of lack of an appropriate antibody) yet some unsubstantiated statements are made e.g. 'β_3_ -AR dependent cAMP pools are formed in caveolae'. Some evidence (referenced in the manuscript) supports the location of some sGC subunits in buoyant caveolae-containing fractions, and the evidence for caveolar location of eNOS is robust. However, comprehensive evidence to support statements regarding the location of the β_3_ signalosome in caveolae is lacking.

We thank the reviewer for this comment. In the past we have shown β_3_-AR association with caveolae in myocytes transgenically expressing the human β_3_-AR, which allowed us to perform membrane fractionation protein assays as well as a proximity ligation assay using a human specific β_3_-AR and Caveolin 3 antibody. We also currently have a paper in press (Dubois-Deruy E ESCHF 2020, in press) containing further evidence of this association via a PLA colocalization signal between the endogenous rat β_3_-AR and AMPK (itself in caveolae) from rat neonatal cardiac myocytes. In any case we added a remark in the Introduction to mention our previous work on the caveolae and β_3_-AR association: “Recently, overexpression of β_3_-AR in transgenic mice has been shown to protect the heart from catecholamine-induced hypertrophy and remodelling via an eNOS)/soluble guanylyl cyclase (sGC)/cGMP-dependent signalling pathway. The same study showed localization of β_3_-ARs together with eNOS in caveolae-enriched membrane fractions, which had been separated via ultracentrifugation (Belge et al., 2014)”.

11) Figure 6. There seems to be some lack of accuracy in depicting current (and past) data from this group. Why are β1-AR only in crest in healthy myocytes (vs. Nikolaev et al. 2010 Science). Are the PDE2/5 symbols misplaced at the bottom of panel A (surely these degrade cGMP in control cells as shown in Figure 2). The data in Figure 5C suggest that the normal β_3_-AR dependent reduction in cAMP seen in control cells is absent in failing cells (i.e. cAMP lower in control cells in the presence of β_3_-AR stimulation). Yet comparison of panel A and B in Figure 6 suggests that cAMP is lower in failing cells in the presence of β_3_-stimulation. Even given an increase in PDE2 expression/activity in failing cells (not shown explicitly) which could increase degradation of cAMP, how can the data in Figure 5 be reconciled with the cartoons in Figure 6?

We thank the reviewer for spotting the misplaced PDE symbols and have changed the overall schematic to better reflect current and old data (see Figure 7 and its legend).

12) It appears that the differently targeted β_3_-AR stimuli applied to T-tubules vs. Crest membrane regions were performed in different cells, with comparisons between them analyzed with non-paired statistics. This is not explicitly stated but is apparent by the statistical test used (Mann Whitney U). This seems problematic given that the level of FRET response varies from cell to cell, and even accepting IBMX-based normalization between cells, the statistical power to identify quantitative changes would be challenging. If getting both stimuli in the same cell is impossible and unpaired analysis is required, then the ability to properly normalize between cells is critical. The concern with it is raise by the key data shown in Figure 3. While the extent to which β_3_-AR-mediates cGMP at the Crest appears similar between healthy and failing cardiomyocytes, the FRET response at T-tubule declines in the latter group. If this was due to a redistribution of β_3_-AR to the Crest region, would you not expect a rise in the% FRET response there? Is it possible the results can be explained by a decline in β_3_-R levels or signal-coupling in the T-tubules after MI? Data in Figure 3E and 3J should be combined into a 2WANOVA, and a statistical test made for the interaction between heart failure (or not) and location where the agonist was applied.

Unfortunately, a proper normalization with the red-DE5 FRET sensor is not yet established and performing both measurements of crest and T-tubule in one cell could lead to potential desensitization of receptors and alterations in the FRET sensor. We appreciate the reviewer’s suggestion that we use a more stringent statistical test and have now used a mixed ANOVA followed by χ2-test. We also updated the text to better reflect the data of an increased crest signal in heart failure.

13) This study is not the first to examine β_3_ signaling in plasma membrane compartments or its modulation by heart failure (see Trappanese et al., Basic Res Cardiol, 2015), however there are differences in their findings to the current ones that deserve comment. They too found β_3_R expressed in both caveolin enriched and heavy (non-lipid raft) regions shown by direct Western Blot. Their results differ however, in that the distribution in each region did not appear to change in the canine HF model (mitral regurgitation), and signaling seemed only observed if the β_3_R was stimulated in the non-lipid raft compartment. The methods were clearly different, but results should be discussed, particularly in light of Comment 12. Ideally, using a membrane fraction, then Cav3 pull down, and β_3_-AR protein analysis would help confirm if there is indeed a redistribution in this model that was not observed in the canine model reported previously. If there is a real difference, the question then becomes what goes on in human – is it more like rat or dog?

We thank the reviewer and believe the suggested experiment could be of potential value if it were feasible. However, due to the relatively low expression of β_3_-ARs and the current lack of a suitable antibody, which could detect rat β_3_-ARs with appropriate specificity, we have refrained from performing this experiment. We agree that the data from Trappanese et al., 2015, does not match our data exactly, however we assume that this can at least in part be explained by the different pathophysiological courses of heart failure in the two different models. We have shown in the past, that in human a residual cGMP increase was detectable after β_3_-AR agonist application on biopsies from patients with different types of heart disease. (see: Gauthier et al., 1998).

14) There are a number of figures shown where the FRET tracings themselves do not appear to represent the summary data. Examples are in Figure 2 and 5. There are also concerns that for some analysis, very few animals were used, (e.g. Figure 1F, Figure 4—figure supplement 1A, B, Figure 5—figure supplement 1, Figure 5C), and this raises concerns about reproducibility.

The reviewers concern about low animal numbers was noted and additional experiments were performed for figures in which only two animals were used, with the exception of Figure 4—figure supplement 1A, B) in order to increase n numbers and alleviate the concerns about reproducibility. We kept the current traces of FRET experiments, since we believe that the traces are representative of the summary data.

15) Figure 5 is another key figure in the study, looking at potential crosstalk between β_3_-AR-stimulated cGMP and cAMP via PDE2 regulation. The modest decline from β_3_-AR agonist in controls seems to be reproduced in the post MI cells too, though the statistical variance is slightly different so one is borderline significant and the other not. The proper test would be a 2WANOVA, where the impact of heart condition is one variable, and the β_3_-AR application the other. This could be done for the β_3_-AR response, and then for +/- PDE2-I response where β_3_-AR stimulation is present. To support the authors' conclusions, there would need to be a significant interaction.The authors choose forskolin to stimulate cAMP and PDE2-dependent hydrolysis, but this should be contrasted to a β-AR agonist to better support their conclusion (e.g. as depicted in Figure 6).

We thank the reviewer for the suggestions and have performed further experiments to increase n-number and for statistics we used a mixed ANOVA followed by χ2-test to incorporate the variance introduced by the different animals. The differences are still significant. We chose forskolin as our cAMP stimulant as we wished to have a direct comparison between control and heart failure cells. Due to β_1_-AR desensitization in heart failure such a direct comparison would have been complicated, if we had opted to use a direct β-AR agonist.

16) The novelty of the immunofluorescent microscopy data is questioned by prior reports showing similar co-distribution of sGC and Caveolin 3 in normal and failing hearts (e.g. Tsai et al., 2012). There is also a concern that the microscopy is based on freshly isolated cells, whereas the functional data were obtained 48 hours post isolation (time needed to introduce the FRET probes). What is the immunohistology distribution after this 48 hours? Myocytes are known to change the localization of ion channels and various GPCRs during this time period.

We thank the Editor and the reviewers for raising this important point. Cardiomyocytes do indeed remodel but the remodelling of rat cells is not so extensive as to preclude useful experiments. Papers from our group (Gorelik et al. Cardiovascular Research, 2006) and others (Pavlovic et al. Exp Physiol, 2010) have demonstrated that factors such as the organization of the surface of the cardiomyocytes (z-groove ratio) and t-tubule density (rat cells lose only 25% of their t-tubule area over the first 48 hrs). Functional factors are not markedly altered, neither is the capacitance of the membrane or the staining of tubular structures after 48 hours despite the cells looking morphologically quite different due to the rounding of the myocytes ends. Rat cells compare favourably with those isolated from mouse which completely lose their form and function over 48 hrs (Pavlovic et al. Exp Physiol, 2010).

17) The MI model is not described, so it is difficult to know the severity of dysfunction generated. Equally important, the location in the heart from which myocytes cells were isolated post MI – is not noted. This is important; e.g. was this peri-infarct, or from the remote territory? How was that determined? How reduced was the EF generated and what other evidence was there supporting this as heart failure – e.g. elevated filling pressures, pulmonary edema, etc.. What was the sex of the animals?

We thank the reviewer for their observation and have added the respective echocardiography and biometric data as Figure 1.

Additionally, we added the following passage into the Results section:

“Echocardiographic and biometric data were collected and are summarised in Figure 1.”

And the following into the Materials and methods section: “Echocardiographic and biometric data are summarised in Figure 1.”

Animals were all male, we have added this information in the Materials and methods section as follows: “All animals were male.”

18) The scheme shown in Figure 6 is interesting but not really confirmed by the data shown and so remains speculative. This is because Figure 5 does not test the relative signaling involved in the two different compartments (TT vs. Crest), and with cAMP being stimulated coupled to a Cav3/localizing β-AR, versus not.

We appreciate the reviewer’s point, but with the experiment shown in Figure 5 (now Figure 6) we wished to detect if there was an overall potential of β_3_-AR activation to lower and thereby attenuate cAMP levels in control and heart failure. We have slightly changed Figure 6 (now Figure 7) to be less speculative. See Figure 7.

19) In Figure 1B and f the authors show a reduced cGMP generation after Isoproterenol stimulation. However, it is not clear whether this is entirely mediated via β_3_ receptors. The authors should do the similar experiments in the presence of CGP plus ICI (β_1_ and β_2_ adrenoceptor blockers).

We agree with the reviewer that it would benefit our work if we could show the β_3_-AR response only in the MI cells (with inhibiting β_1_ and β_2_ receptors); however, we were unable to perform the suggested experiment as we did not have any more MI cells (in the 2 months of time that we were given for the revision).

20) In Figure 2E it is not entirely clear whether the FRET response by IBMX (about 4%) represents the cGMP generation on top of Isoproterenol evoked cGMP generation. If not the IBMX effect together with Isoproterenol is not larger than with ISO alone (Figure 1F). This should be clarified.

We thank the reviewer for this point and have tried to make the data description clearer by adding the following passage into the figure legend: “The scatter plot/histograms present the average whole cell cGMP-FRET responses evoked by PDE inhibition further to the isoproterenol responses in% from (A-D) (E).”, that means i.e. if the ratio after ISO application was at 0.98 and the following ratio after single PDE inhibition was at 0.97 and the ratio after IBMX application at 0.96, then the values would be 1 for the single PDE and 2 for the total PDE inhibition.

21) In the first paragraph of the subsection “Functional β_3_-ARs are localized in the T-tubules of healthy cells and redistribute to the non166 tubular sarcolemma in heart failure”, the authors state that in failing hearts cGMP generation can be detected across the sarcolemma. The authors should more specifically state that in healthy cardiomyocytes there is no crest response which becomes now obvious in failing hearts. At least this is demonstrated in Figure 3D and I.

We thank the reviewer for this comment and have changed the manuscript text accordingly to “In healthy cardiomyocytes, we observed that functional β_3_-ARs reside mainly in the T-tubules with very few responses being detectable outside of T-tubules (Figure 4A-E), whereas in failing cells, they β_3_-ARs responses after localised stimulation can be detected in both tubulated and non-tubulated areas across the sarcolemma (Figure 4F-J).”

22) The immunostainings shown in Figure 4 and Figure 5—figure supplement 1 entirely depend on the specificity of the antibodies directed against sGCα1, caveolin3 and sGCβ1. The authors should comment in the manuscript about what is known about the specificity of these antibodies. Have they been used previously in knockout control cells or what other efforts have been made to demonstrate their specificity?

We agree with the reviewer that the specificity of antibodies is essential for the experiments shown in our manuscript. Therefore, we used the best available antibodies and added the specificity info into the text. We have commented on the specificity of the antibodies we employed in our manuscript in the Materials and methods section as follows: “rabbit sGCα and β subunit antibodies (Prof. Andreas Friebe from Würzburg, Germany, specificity tested in KO animals Friebe, Voußen and Greoneberg, 2018), mouse α-actinin (Σ Aldrich, A7732), mouse Caveolin 3 (BD Transduction Laboratories, 610421, specificity tested in KO animals (Woodman et al., 2002)”.

Additionally, we tested the specificity of sGCα. Freshly isolated adult mouse cardiomyocytes (either from wild type of from global sGC knockout mice) were fixed with ice-cold methanol and co-stained with sGCb1 and Cav3 antibodies as described in the Materials and methods and Figure 5—figure supplement 1) Confocal images were using the small laser intensities in all conditions similar to those used in this manuscript.

23) Figure 4—figure supplement 1: In the model in figure B and C, it is suggested that cGMP formation is reduced in the T tubules after induction of heart failure. However, if there is no statistical significance in Figure 4—figure supplement 1A, B between the first bar (T tubule scramble) and the third bar (T tubular C3SD). Thus, this model is not supported entirely by the data.

We thank the reviewer for the observation and have altered the cartoon accordingly, so that the cGMP formation in HF tubules does not appear smaller. See figure 4—figure supplement 1.

24) In Figure 5 the authors should somehow indicate more clearly at which time points the changes in cAMP generation are quantified in C. Is this at the time point just before IBMX application? The authors should clearly state in the manuscript the percentage of inhibition in cAMP generation by the β_3_-agonists. It seems to be in the range of 10-15%?

We thank the reviewer for the suggestion and have attempted to clarify the measurement steps in the manuscript with the following explanation in the Figure 5 figure legend: “Scatter plot/histogram presenting whole cell cAMP-FRET responses depicted as the percentage of the maximal possible cAMP FRET response (= Forskolin followed by IBMX). The measured Forskolin or IBMX responses were the respective maximal responses, equalling the lowest FRET ratio value, achieved after each stimulus.” Additionally, we added the average response rate of 10.3% into the manuscript text as follows: “In healthy cells, stimulation of β_3_-AR led to a significant reduction of approximately 10.3% of the forskolin stimulated cAMP production (Figure 5C)”.

25) The model in Figure 6 in healthy cardiomyocytes indicates that cGMP act on PDE2 to limit cAMP generation. The authors should elaborate in a little bit more detail how this is achieved for readers not entirely familiar with the previous literature.

We thank the reviewer for the suggestion and have elaborated more on this point in the Discussion section as follows: “At the same time as being degraded by PDE2, cGMP can also increase PDE2 activity on cAMP more strongly by binding to it, thereby triggering the so-called cGMP-to-cAMP crosstalk.This crosstalk between the second messengers of which one (cAMP) has increasingly been associated with detrimental signalling pathways in the context of heart failure, which the other (cGMP) could potentially attenuate (Mongillo et al., 2006; Moniotte et al., 2001; Yanaka et al., 2003)”, as well as in the figure legend (see Figure 7).